# Multiple Phosphorylations of SR Protein SRSF3 and Its Binding to m^6^A Reader YTHDC1 in Human Cells

**DOI:** 10.3390/cells11091461

**Published:** 2022-04-26

**Authors:** Takanori Tatsuno, Yasuhito Ishigaki

**Affiliations:** Medical Research Institute, Kanazawa Medical University, Kahoku 920-0293, Japan; ttntknr@gmail.com

**Keywords:** N6-methyladenosine, YTHDC1, SRSF3, phosphorylation

## Abstract

N6-methyladenosine (m^6^A) is a well-known RNA modification and has various functions with its binding proteins. Nuclear m^6^A reader protein YTHDC1 plays a significant role in RNA metabolism including some non-coding RNA such as LINE or circRNA. It is also known to regulate mRNA splicing through recruiting SRSF3 to the targeted mRNAs, which then mediates export of YTHDC1-bound RNA to the cytoplasm. Additionally, it has been indicated that SRSF3 binding to YHTDC1 may be mediated by its dephosphorylated status. However, their binding mechanism, including the positions of dephosphorylated residues of SRSF3, has not been sufficiently investigated. Thus, we explored the mechanism of interaction between SRSF3 and YTHDC1 in human cells. We used co-immunoprecipitation to examine the binding of YTHDC1/SRSF3 through their N- and C-terminal amino-acid residues. Furthermore, dephosphorylation-mimic serine to alanine mutants of SRSF3 indicated the position of phosphorylated residues. Cumulatively, our results demonstrate that YTHDC1 binding to SRSF3 is regulated by not only hypo-phosphorylated residues of arginine/serine-rich (RS) domain of SRSF3 but also other parts of SRSF3 via YTHDC1 N- or C-terminal residues. Our results contribute to the understanding of the complex mechanism of binding between SR protein SRSF3 and the m^6^A reader YTHDC1 to regulate the expression of mRNA and non-coding RNAs.

## 1. Introduction

N6-methyladenosine (m^6^A) is one of the most investigated modifications of eukaryotic mRNAs and is related to various mRNA metabolisms, such as RNA decay, splicing, transport, and translation efficiency [1,2,3,4,5,6,7,8,9]. The m^6^A modification is also potentially important in miRNA biogenesis, long non-coding RNA expression, RNA structures, and circular RNA (circRNA) biogenesis; however, the mechanisms of these physiological functions, including the relevant binding proteins, have not been investigated sufficiently [10,11,12,13,14].

In m^6^A regulation, there are three categories of proteins involved: “writers” (m^6^A methyltransferases), “erasers” (m^6^A demethyltransferases) and “readers” (effectors recognizing m^6^A). Recent studies have identified multiple components belonging to these categories. METTL3 and METTL14 are core components of “writers” and add methyl residues at position N6 of adenosine, together with the accessory components WTAP, VIRMA, RBM15, and ZC3H13 [1,15,16,17]. The methyl residues are reversibly removed by “erasers”, which include FTO and ALKBH5 [2,18]. “Readers” recognize m^6^A by the YT521-B homology (YTH) domain, and subsequently direct RNA metabolism. In addition to YTH domain family proteins, other proteins, including several heterogeneous nuclear ribonucleoproteins, have been known to work as “readers” [10,19]. The YTH domain-containing family consists of YTHDF1, YTHDF2, YTHDF3, YTHDC1, and YTHDC2 and each of these has different physiological functions [20]. For instance, YTHDF2 recruits the deadenylase complex CCR4-NOT and degrades RNA in mammalian cells [21], whereas YTHDF1 interacts with translation initiation factors to regulate mRNA translation [4].

Among YTH domain-containing family proteins, only YTHDC1 localizes in the nuclear compartment and has significant roles in splicing, export of RNA, reshaping nuclear speckles with MALAT1, or regulation of scaffold function of LINE1 [22,23,24,25]. Serine-arginine (SR) proteins are necessary to work with YTHDC1 in these mechanisms. YTHDC1 is known to interact with SRSF1 (ASF/SF2), SRSF3 (SRp20), SRSF7 (9G8), SRSF9 (SRp30c), and SRSF10 (SRp38). Among interacting SR protein, SRSF3 is involved in exon inclusion of YTHDC1-binding methylated RNAs and its localization to nuclear speckles is affected by YTHDC1 [22].

As for SRSF3, it has been reported that SRSF3 has great influence on various type of cancers via splicing regulation [26,27,28,29] In addition, dephosphorylated SRSF3 selectively mediates export of methylated mRNAs to the cytoplasm through YTHDC1 binding by association with NXF1 [23]. Although several SR proteins can be phospho- or dephosphorylated, the role or the positions of SR protein phosphorylation remains poorly understood. Furthermore, despite evidence for interaction between YTHDC1 and SRSF3, the detailed mechanisms of binding have not been completely revealed. Here, we demonstrate the position of SRSF3 phosphorylation sites and detail the amino acid sequences needed for the binding between SRSF3 and YTHDC1.

## 2. Materials and Methods

### 2.1. Cell Culture

The HeLa cell line was acquired from RIKEN Bioresource Center and was cultured in Dulbecco’s modified Eagle medium (Wako Pure Chemical Industries, Osaka, Japan), with 4.5 g/L glucose supplemented with 10% fetal bovine serum (Sigma-Aldrich, St. Louis, MO, USA), 100 mg/mL penicillin, and 100 U/mL streptomycin (Wako) at 37 °C in 5% CO_2_.

### 2.2. Vector Construction

SRSF3 or YTHDC1 expression vectors were constructed from the pReceiver vector (Genecopoeia) and PCR amplified SRSF3 and YTHDC1 sequences from extracted Hela mRNA. The SRSF3 serine-to-alanine mutant vector was constructed as previously described [30,31]. Each mutation or deletion was generated using PrimeSTAR Max DNA Polymerase (TaKaRa Bio, Kusatsu, Japan) or the QuikChange Site-Directed Mutagenesis Kit (Agilent Technologies, Santa Clara, CA, USA). Primers were obtained from IDT and their sequences are provided in Table 1. Primers used for SA mutations were generated by replacing the first two bases in the serine codon with GC and generated according to manufacturer’s instructions. An mCherry-tagged SRSF2 vector was made from the pReceiver vector with mCherry-SRSF2 custom coding using DNA fragments (GeneArt string fragments, Thermo Fisher Scientific, Waltham, MA, USA). Constructed vectors were sequenced with BigDye Terminator v3.1 Cycle Sequencing Kit (Thermo Fisher Scientific, Waltham, MA, USA) and a 3500xL Genetic Analyzer (Thermo Fisher Scientific, Waltham, MA, USA).

### 2.3. Co-Immunoprecipitation

Co-immunoprecipitation was performed as previously described [22] with minor modifications. HeLa cells were transfected with indicated pReceiver plasmids using Lipofectamine 3000 (Thermo Fisher Scientific, Waltham, MA, USA). Cells were harvested after 48 h of transfection with lysis buffer (20 mM Tris-HCl pH 7.4, 150 mM NaCl, 0.5% NP-40, 1 × Protease inhibitor cocktail (Roche, Basel, Switzerland)). The lysate was sonicated at 10% power (6 cycles, pulse-on: 10 s, pulse-off: 20 s) and cleared by centrifugation at 15,000× *g* for 15 min. The supernatant was incubated with RNase A (1 mg/mL) (Sigma, St. Louis, MO, USA) for 40 min at 20 °C and subsequently cleared by centrifugation. The lysate was then incubated with rabbit anti-Myc or anti-GFP antibodies (MBL, Tokyo, Japan) with gentle rotation overnight at 4 °C. Agarose G beads (Invitrogen, Waltham, MA, USA) were added and rotated for 2 h at 4 °C. Beads were washed with a lysis buffer six times to remove non-specific proteins. After washing, beads were boiled in 2 × SDS loading buffers. The immunoprecipitants were analyzed by SDS-PAGE/western blot analysis.

### 2.4. SDS-PAGE/Western Blot Analysis

To elucidate the phosphorylation status, we used phos-tag SDS-PAGE, which separates phosphorylated and non-phosphorylated proteins, based on their levels by capturing phosphate residues. Phos-tag gels were self-made using Phos-tag acrylamide reagent (Wako, Monza, Italy). Cell lysates or precipitates from co-immunoprecipitation assays were used for western blotting. Each sample was mixed with a Laemmli sample buffer (Wako, Monza, Italy), resolved on SDS-PAGE or phos-tag SDS-PAGE, and transferred to PVDF membranes. Protein-bound membranes were blocked with a 5% skim milk buffer and incubated with a primary antibody diluted in 0.5% skim milk buffer. Mouse anti-Myc antibody (1:2000, MBL) and mouse anti-GFP antibody (1:3000, Santa Cruz Biotechnology, Santa Cruz, CA, USA) were used as primary antibodies. Horseradish peroxidase-conjugated anti-mouse IgG antibody was used as the secondary antibody (Agilent Technologies, Santa Clara, CA, USA). The membrane was subsequently washed and developed with ImmunoStar Zeta chemiluminescent reagents (Wako, Monza, Italy) and chemiluminescence was visualized using the Fusion (Vilber-Lourmat, Collégien, France).

## 3. Results

### 3.1. Analysis of GFP-Tagged SRSF3 and Fragments

We first focused on the arginine/serine-rich (RS) domain of SRSF3 as this contains two serial RS repeats similar to those of SRSF1 (Figure 1a). As with SRSF1, these two RS repeats are known to be phosphorylated sequentially depending on their localization or binding stages [32,33]. Therefore, to analyze the phosphorylation pattern of the RS domain on SRSF3, we constructed green fluorescent protein (GFP)-tagged SRSF3 and partial deletion mutants covering SRSF3 regions (1–83), (84–164), (84–132) and (133–164) (Figure 1b). The SRSF3(1–83) mutant, which lacks the RS domain, was devoid of RS repeats but retained the RNA recognition motif (RRM, River Vale, NJ, USA). Conversely, SRSF3(84–164) lacked the RRM and only contained the RS domain, resulting in a loss of RNA binding. SRSF3(84–132) and SRSF3(133–164) contained the first RS repeats and the latter RS repeats of the RS domain, respectively. Firstly, the expression of these proteins was confirmed by western blotting (Figure 1c, left).

Next, their phosphorylation states were elucidated by western blotting using phos-tag gel SDS-PAGE (Figure 1c, right). The phos-tag analysis confirmed that there was no phosphorylation site within the RRM (River Vale, NJ, USA), because band shift was not detected in RRM fragment of SRSF3. However, the band shifts in phos-tag gel indicated the strong phosphorylation of RS domain and this result is consistent with previous report [34]. Moreover, the phosphorylation of SRSF3(84–132) and SRSF3(133–164) was detected, although these fragments did not consist of a complete RS domain.

### 3.2. Phosphorylation of RS Domain of SRSF3

We next explored the position of serine phosphorylation on the RS domain of SRSF3 as previously described in [30]. SRSF3 phosphorylation was implied to occur across the entire RS domain based on the fragment analysis. We constructed Myc-SRSF3 expression vectors with serine to alanine (SA) mutations at almost all the serine positions in the RS domain, although we could not construct an SA mutant at position 130.

From the phos-tag analysis (Figure 2a), results showed that SA mutants at positions 108, 115, and around 124–130, produced lower band shifts. We also observed that although the phosphorylation pattern of the SA mutants at positions around 124–130 was similar to that of the N-terminal SRSF1 phosphorylation RS repeats by SRPK1 [35], many SA mutations residing at the latter half of the RS domain showed upper band shifts. As a result, the replaced alanine residues could not be phosphorylated (Figure 2b). Additionally, multiple band shifts showed the phosphorylation of multiple serine residues in SRSF3 polypeptide.

To reveal the multiple phosphorylation status of SRSF3, we then examined the phosphorylation status with multiple replacement mutants (Figure 3a). As a result, the 815SA mutant (SA mutation at positions 108 and 115), 2430SA mutant (SA mutation at positions 124, 126, 128, and 130), and MSA1 mutant (SA mutation at positions 108, 115, 124, 126, 128, and 130) had different patterns of band shift which indicated both of these sites were phosphorylated (Figure 3b). The 5460SA mutant (SA mutation at positions 154, 156, 158, and 160) had a hyperphosphorylated band shift similar to single mutants at C-terminal RS domain, whereas the MSA2 mutant (SA mutation at positions 108, 115, 124, 126, 128, 130, 154, 156, 158, and 160) exhibited a band shift similar to the 815 and 2430SA mutant.

Based on phos-tag gel electrophoresis, it was suggested that phosphorylation occurred on SRSF3, at positions 108, 115, and near 124–130. Furthermore, hyperphosphorylation from replacement at positions 154–160 disappeared when N-terminal phosphorylated serine residues were replaced with alanine. Thus, the replacement of the C-terminal serine residues implied that the hyperphosphorylated site existed at positions 108, 115, and/or 124–130. Therefore, we concluded that endogenous SRSF3 was phosphorylated at positions 108, 115, and near 124–130.

### 3.3. Binding Ability of SRSF3 with YTHDC1

Next, to verify the SRSF3 binding sites of YTHDC1 (Figure 4a), we performed a co-immunoprecipitation assay with Myc-tagged YTHDC1. Expression vectors of GFP-tagged SRSF3 and Myc-tagged YTHDC1 were co-transfected into HeLa cells, and GFP-tagged SRSF3 mutants were detected by western blotting after co-immunoprecipitation with anti-Myc antibody. Unexpectedly, not only the RS domain but also the RRM were precipitated by YTHDC1 unlike previously published results (Figure 4b left, phos-tag result as shown in Figure 4b right) [22].

Therefore, we examined SRSF3 binding with the C-terminal fragments of YTHDC1 (Figure 5). We predicted that even though only the RRM region coimmunoprecipitated with the C-terminal YTHDC1 fragment, YTHDC1(493–727) (containing arginine-rich amino acid sequences), Myc-YTHDC1 was also coimmunoprecipitated. These results imply that YTHDC1 was bound to the RRM and RS domains of SRSF3 through C-terminal sequences.

### 3.4. Interaction of the N-Terminal Half of RS Domain of SRSF3 with the Glutamic Acid-Rich N-Terminal Region of YTHDC1

These results posed questions concerning how the described N-terminal YTHDC1 binding to RS domain of SRSF3 had an effect on the interaction between YTHDC1 and SRSF3, or which sites of RS domain were used for binding. Thus, to clarify these questions, we generated constructs comprising the SRSF3 RRM with the N-terminal or C-terminal half of RS domain, SRSF3(1–132) or SRSF3(Δ84–132), respectively. In addition, we also constructed an SA mutant, as for SRSF3(1–132), where mutations were inserted at positions 108, 115, and 124–130.

From the results of co-immunoprecipitation and western blotting, we confirmed YTHDC1 binding to SRSF3(1–132) and the SA mutant (Figure 6a). The 1-132SA mutant 247 is 1–132 SRSF3 with 108, 115, and 124–130SA mutations However, the N-terminal RS domain deletion mutant SRSF3(Δ84–132) interacted less with YTHDC1 compared with SRSF3(1–132). In addition, both SRSF3(1–132) and SRSF3(Δ84–132) were confirmed to be dominantly phosphorylated from the phos-tag gel analysis and shown to bind to YTHDC1 with hypo- or de-phosphorylated status. Next, co-immunoprecipitation was performed with YTHDC1(1–353) to evaluate N-terminal YTHDC1 binding with the RS domain. YTHDC1(1–353) strongly interacted with SRSF3(1–132) and slightly bound to full-length SRSF3, although there was no interaction between SRSF3(Δ84–132) and YTHDC1(1–353) (Figure 6c). This result differs from that of Roundtree et al. [23], i.e., that the N-terminus of YTHDC1 had no binding ability, which may be due to differences in experimental conditions such as buffer and the effect of the tag.

Investigations also confirmed the binding of YTHDC1(1–353) to the SRSF3(1–132) SA mutant (Figure 7). However, our results showed no relationship between phosphorylation and binding to YTHDC-1.

## 4. Discussion

Following the identification of m^6^A reader proteins, various associators of YTH proteins have been found to be related to processes in RNA metabolism, such as mRNA translation or RNA stability. However, as initial investigations focused on YTHDF2 in the cytoplasm, the described physiological functions of the YTH proteins have mainly been cytoplasmic [3]. Thus, the functional mechanism of the nuclear m^6^A binding protein YTHDC1 remains largely unknown. However, YTHDC1 is known to affect alternative splicing [22,36] and has potential for the nuclear transport of mRNAs [23], and several of these YTHDC1 functions have been associated with the SR protein SRSF3. Although splicing and nuclear export of mRNA are related to each other, their underlying machinery has been separately investigated to some extent, as with SRSF3 and YTHDC1. Previous reports revealed that the RS domain of SRSF3 appeared sufficient for binding during splicing or nuclear export [22,23], whereas the present study showed that the RRM region of SRSF3 could bind to YTHDC1 and thereby support the machinery of splicing or nuclear export. Furthermore, the first half of the RS domain of SRSF3 was proven to be essential for phosphorylation of SRSF3 and tight association with YTHDC1.

In this study, we focused on the binding between SRSF3 and YTHDC1. Previously, the RS domain of SRSF3 was reported to be phosphorylated [34]; the phosphorylation or dephosphorylation of SR proteins led to the respective association or dissociation with spliceosomal factors [37]. The dephosphorylated SRSF3 in messenger ribonucleoprotein particles (mRNPs) may facilitate their export with YTHDC1 interaction [23]. We first determined the phosphorylation sites using fragments of SRSF3 and phos-tag gel as the locations of phosphorylated serine residues within SRSF3 were unknown in vivo, although the SRPK2-mediated phosphorylated site is confined throughout its entire RS domain in vitro [38,39]. This analysis implied that the RS domain of SRSF3 was phosphorylated, whereas the RRM domain lacked any phosphorylation. In addition, both the N-terminal and C-terminal halves of the RS domain had up-shifted bands in phos-tag gel analysis. Thus, it was concluded that phosphorylation could occur throughout the RS domain.

Next, we investigated the phosphorylated serines in succession using serine to alanine replaced mutants. Based on phos-tag gel electrophoresis, it was supposed that phosphorylation was induced at positions 108, 115, and near 124–128. Unexpectedly, the SA mutants in the C-terminal half of the RS domain displayed hyperphosphorylation. Therefore, at least, the C-terminal half of the RS domain was not supposed to be phosphorylated, but different from result of C-terminal RS domain fragment.

Subsequent phos-tag analysis with multiple replacement mutants specifically identified the phosphorylated serine residues and demonstrated that phosphorylation of SRSF3 occurred only within the N-terminal RS domain, and hyperphosphorylated residues came from the C-terminal SA replacement and existed at the N-terminal half of the RS domain. Therefore, it was suggested that the C-terminal RS domain had potential to be phosphorylated but phosphorylation was restricted by the N-terminal half of the RS domain, and might have another role, such as a scaffold for CLK1, as in the case of SRSF1, in regulating phosphorylation in the N-terminal half of the RS domain [40]. In relation to the binding or function between SRSF3 and YTHDC1, previous studies used YTHDC1 fragments from residues 1–353 or 493–727 [22,23]. The YTHDC1(1–353) fragment was shown to associate with RS domain, whereas the YTHDC1(493–727) fragment was shown to affect transport of mRNA under SRSF3 existence.

The RS domain was successfully co-immunoprecipitated with Myc-tagged YTHDC1, consistent with previously published data [22]. Unexpectedly, we also observed a band of the SRSF3 RRM via Myc-tagged YTHDC1 co-immunoprecipitation. This may have come from using a GFP-tag to SRSF3 or its fragments for co-immunoprecipitation. Since GFP was able to diffuse throughout the cell without any fusion protein, GFP-tagged proteins might result in different localization compared with that of other tags.

Surprisingly, the results of co-immunoprecipitation with YTHDC1(493–727)-truncated protein resembled that of the full-length YTHDC1 protein, indicating that the C-terminal region of YTHDC1, which had an arginine-rich sequence, may bind to SRSF3, even though this lacks a methylated RNA-binding YTH domain. Although the results from the immunoprecipitation of YTHDC1(493–727) were unexpected, it might be possible since the arginine-rich sequence of YTHDC1 contains a phosphorylation that mimics RS repeat-like sequence, SRPRERDRERERD, and it has been reported that some of the RS domains can interact with each other [41].

In addition, the interaction between RRM and RS domain or RS repeats like sequence is plausible since SRSF1 shows an intra-molecular interaction between RRM and RS domain [42]. The full-length YTHDC1 bound with SRSF3(1–132) or its SA mutant, whereas SRSF3(Δ84–132) was only slightly precipitated. A co-immunoprecipitation assay with YTHDC1(1–353) demonstrated that this bound strongly to SRSF3(1–132) and weakly interacted with the full-length SRSF3, but there was no interaction with SRSF3(Δ84–132). These results might be artificial to some extent since the deletion affected phosphorylation status of the mutants. However, the amount of YTHDC1(1–353)-bound hypo- or de-phosphorylated SRSF3 mutants were clearly different, even though both SRSF3(1–132) and SRSF3(Δ84–132) were dominantly phosphorylated. Thus, it was indicated that the N-terminal site of the SRSF3 RS domain might play a major role in YTHDC1 binding.

Additionally, phosphorylated SRSF3 was precipitated to a minor extent by co-immunoprecipitation by YTHDC1 (Figure 4b, right panel). Furthermore, co-immunoprecipitation with deletion SRSF3 mutants revealed that hypo-phosphorylated SRSF3(1–132) bound to YTHDC1. Moreover, YTHDC1(1–353) could bind to dephosphorylated SRSF3 deletion mutant SRSF3(1–132) SA (Figure 7). These results indicated that the YTHDC1 interaction with SRSF3 was not as straightforward as the N-terminal glutamic acid-rich region of YTHDC1, which could bind to the N-terminal site of the SRSF3 RS domain, whereas the C-terminal arginine-rich region of YTHDC1 interacted with both the RRM and RS domains of SRSF3. Together with previously published results [22], this suggests that the N-terminal region of YTHDC1 binds with SRSF3 via the N-terminal region of the hypo- or de-phosphorylated SRSF3 RS domain, resulting in binding to mRNA that contains m^6^A with an adjacent SRSF3 binding site and subsequent RNA splicing. As the binding conformation with SRSF3 was shown to be different between N-terminal YTHDC1 and C-terminal YTHDC1 regions, m^6^A bound YTHDC1 may direct mRNA-binding hypo- or de-phosphorylated SRSF3, which has already completed the splicing reaction and reside at proper position for NFX1 binding, resulting in mRNA export.

Several studies have reported the difference in YTHDC1 expression in a variety of cancers [33,43,44]. Although the altered expression of YTHDC1 has been linked with m^6^A modification machinery and the effect on oncogenic factors, such as BRCA2 or PGR [45], it has yet to be addressed how YTHDC1 affects splicing or export in cancer cells. Recently, it was proven that the circRNA circ-ZNF609, which requires YTHDC1 for its back-splicing reaction, increases in pathological conditions, such as rhabdomyosarcoma [14]. Thus, YTHDC1 can affect not only mRNA but also other RNA, including circRNA, in cancer. In addition, SRSF3 or other SR proteins are known to associate with various cancers [26,27,28,29,46]. However, little is known about the splicing or export machinery of m^6^A-deposited RNA by SR protein, and not all SR protein expression is reported to be controlled by m^6^A modification. With the exception of SRSF1, the phosphorylated sites or mechanisms for SR proteins are mostly unreported. Therefore, it is possible that part of the proteomic diversity in cancer cells may be regulated by unknown mechanisms that derive from m^6^A- deposited RNA splicing or export with YTHDC1 or SR proteins, including their various phosphorylation statuses.

In conclusion, the RS domain of SRSF3 plays a role in both splicing and nuclear export via interactions with the N- or C-terminal sequence of YTHDC1. In addition, the SRSF3 RS domain and RRM domain were revealed to interact with YTHDC1 through its C-terminal region. We have clarified the phosphorylation sites of SRSF3 and provided a mechanistic basis for the binding between YTHDC1 and SRSF3. Previous reports did not indicate a preference in splicing regulation with YTHDC1-interacting SR proteins, such as SRSF1 or SRSF7, since the phosphorylation status of the RS domain or similar sequences was not completely elucidated; therefore, further investigation should be performed to reveal the regulation of m^6^A by SR proteins or proteins that possess RS repeats.

## Figures and Tables

**Figure 1 cells-11-01461-f001:**
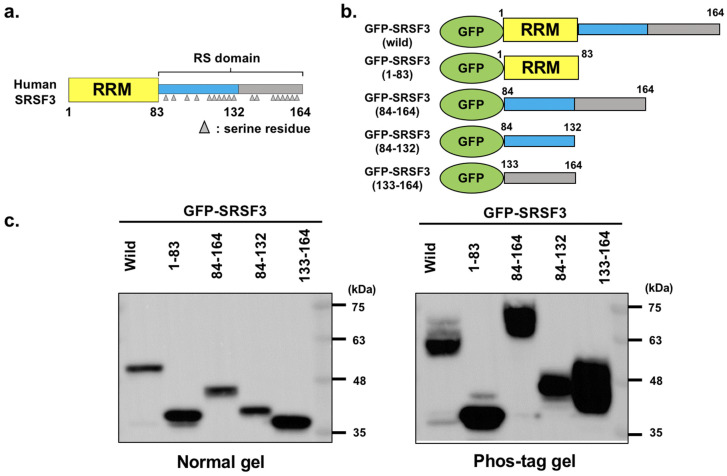
SRSF3 structure and the estimation of SRSF3 fragments. (**a**) SRSF3 structure indicating the position of RNA-recognition motif (RRM), RS domain (the first half of RS domain containing RS repeats is shown in the blue box, the latter half of RS domain containing RS repeats is shown in the gray box), and serine residues (gray triangles). (**b**) Structure of SRSF3 and fragment mutants. (**c**) Expression of transfected GFP-SRSF3 and fragment mutants was detected by western blotting with an anti-GFP antibody (left panel). Phosphorylation status of SRSF3 and mutants was analyzed with phos-tag gel (right panel).

**Figure 2 cells-11-01461-f002:**
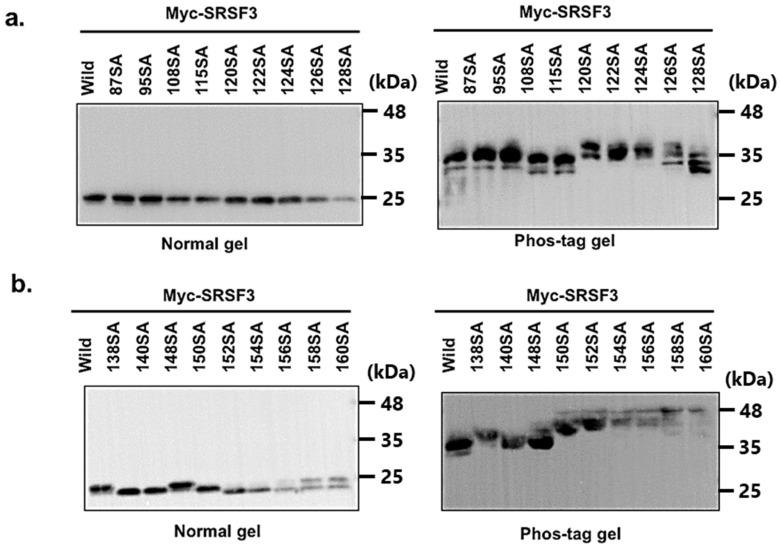
Analysis of phosphorylation status of SRSF3 with single or multiple serine to alanine replaced mutants. 87–128 mutants (**a**) and 138–160 SA mutants (**b**). Expression of transfected Myc-SRSF3 and single serine to alanine replaced mutants was detected by western blotting with an anti-Myc antibody (upper left panel, mutated serine residues derived from N-terminal half of RS domain; lower left panel, mutated serine residues derived from C-terminal half of RS domain). Phosphorylation status of SRSF3 and mutants was analyzed by separation using phos-tag gels (right panel).

**Figure 3 cells-11-01461-f003:**
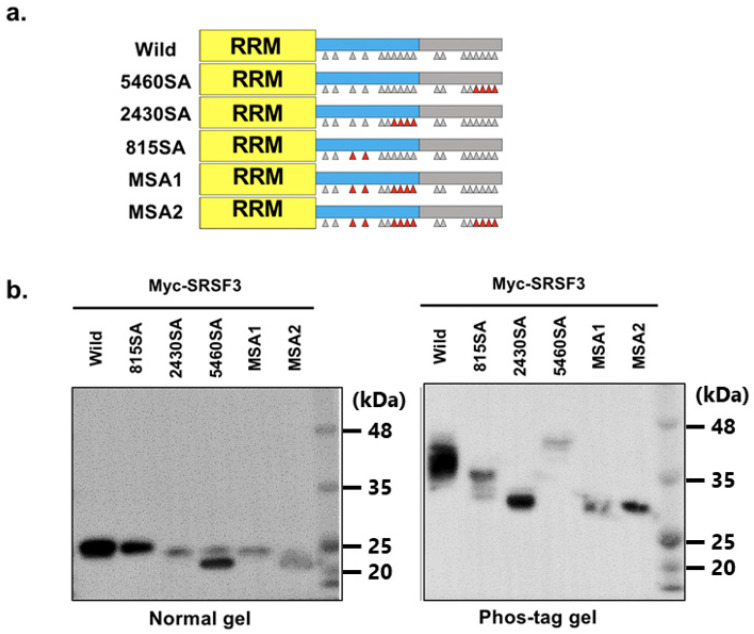
Analysis of phosphorylation status of SRSF3 with multiple serine to alanine replaced mutants. (**a**) Structure of SRSF3 multiple serine to alanine replaced mutants. Replaced serine are shown as a red triangle. (**b**) Expression of transfected Myc-SRSF3 and multiple serine to alanine replaced mutants was detected by western blotting with an anti-Myc antibody (left panel, separation with SDS-PAGE; right panel, separation with phos-tag SDS-PAGE).

**Figure 4 cells-11-01461-f004:**
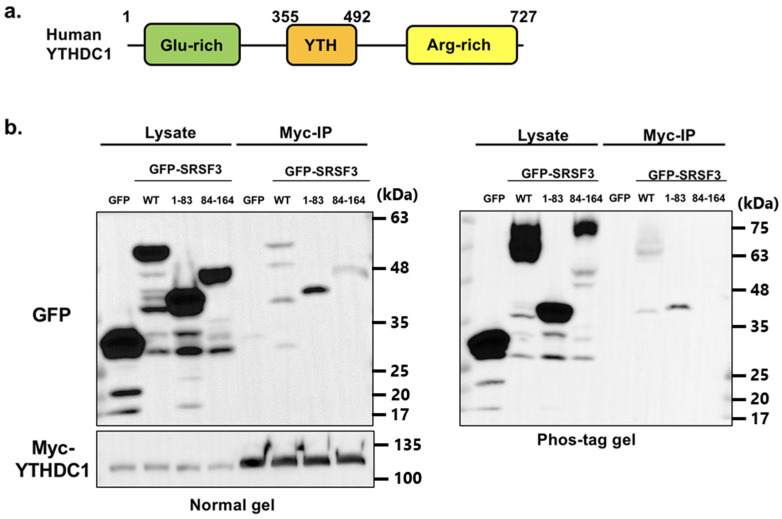
Interaction validation of YTHDC1 with SRSF3. (**a**) YTHDC1 and its fragment structure indicating the position of glutamic acid-rich region (Glu-rich), YT521-B homology (YTH) domain and arginine-rich region (Arg-rich). (**b**) HeLa cells were co-transfected with GFP-SRSF3, Myc-YTHDC1, or their mutants. The lysates of co-transfected cells were immunoprecipitated using anti-Myc antibody. Myc-YTHDC1 interaction with GFP-SRSF3 and its fragments was detected by western blotting using anti-GFP or anti-Myc antibodies ((**b**), **left**); the result of phos-tag gel is also shown in ((**b**), **right**). Some nonspecific bands are observed.

**Figure 5 cells-11-01461-f005:**
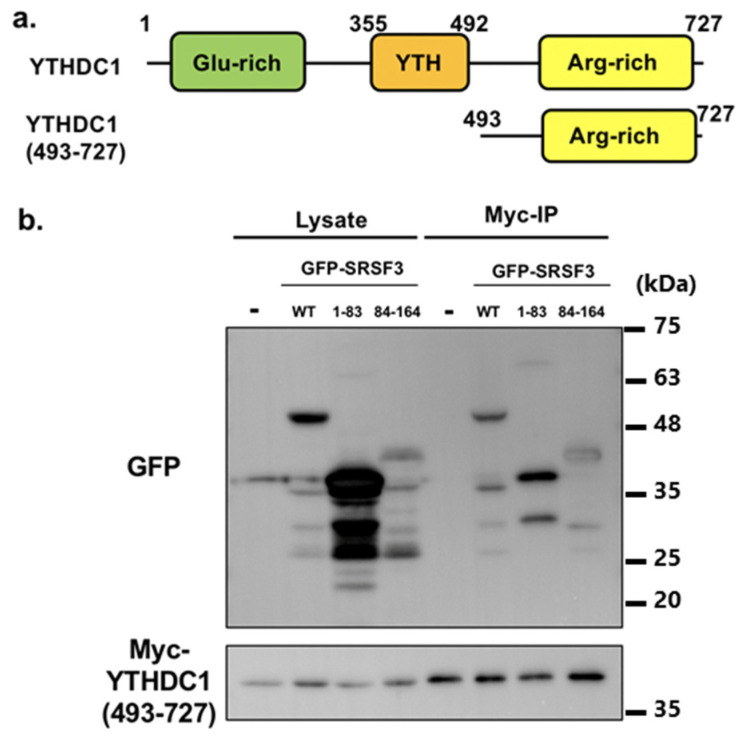
Interaction of YTHDC1 fragment with SRSF3. (**a**) YTHDC1 and its 493–727 fragment structure. (**b**) HeLa cells were co-transfected with GFP-SRSF3 wild type and mutants, and Myc-YTHDC1(493–727). The interaction between the C-terminal region of YTHDC1 with GFP-SRSF3 and its fragments was detected by western blotting using anti-GFP or anti-Myc antibodies. Some nonspecific bands are observed.

**Figure 6 cells-11-01461-f006:**
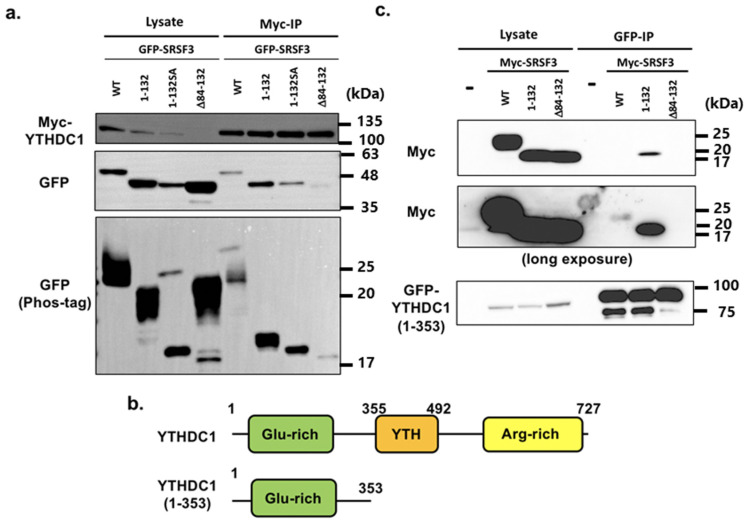
(**a**) HeLa cells were co-transfected with Myc-YTHDC1, GFP-SRSF3, or GFP-SRSF3 deletion mutants. The lysates of co-transfected cells were immunoprecipitated using anti-Myc antibodies. Lysates and IP samples were detected by western blotting using anti-GFP or anti-Myc antibodies. Phosphorylation status of immunoprecipitated SRSF3 and mutants was analyzed by separation using phos-tag gel (bottom panel). (**b**) YTHDC1 and its 1–353 fragment structure. (**c**) HeLa cells were co-transfected with GFP-YTHDC1(1–353), Myc-SRSF3, or Myc-SRSF3 deletion mutants. Lysates of co-transfected cells were immunoprecipitated using anti-GFP antibody. Lysates and IP samples were detected using anti-GFP or anti-Myc antibodies. YTHDC1(1–353) shows a clear interaction with SRSF3(1–132).

**Figure 7 cells-11-01461-f007:**
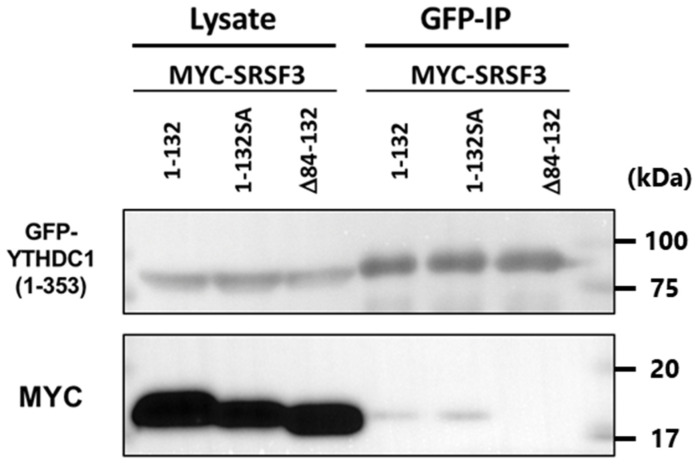
HeLa cells were co-transfected with GFP-YTHDC1(1–353), Myc-SRSF3, or Myc-SRSF3 deletion mutants. The lysates of co-transfected cells were immunoprecipitated using anti-GFP antibody. Lysates and IP samples were detected by western blotting using anti-GFP or anti-Myc antibodies.

**Table 1 cells-11-01461-t001:** Primers used in this study.

Primer Name	Primer Sequence (5′→3′)
YTHDC1(1−)-F	AGCGTTCGAACCATGGCGGCTGACAGTCGG
YTHDC1(−727)-R	GCCGCACTCGAGCTATCTTCTATATCGACCTCTC
YTHDC1(−353)-R	GCCGCACTCGAGCTATTGAAGCACATATTTGAG
YTHDC1(493−)-F	AGCGTTCGAACCCCCCCCGATGAAAGTATTGAC
SRSF3(1−)-F	AAGTCTTTCGAACCATGCATCGTGATTCCTGTCC
SRSF3(84−)-F	AAGTCTTTCGAACCGAAAAAAGAAGTAGAAATCG
SRSF3(133−)-F	AAGTCTTTCGAACCAGGAGAAGAGAGAGATCGCTG
SRSF3(−83)-R	GTTCTTGCGGCCGCCTAACCATTCGACAGTTCCAC
SRSF3(−132)-R	TCTTGCGGCCGCCTAATCTCTAGAAAGGGACCTGC
SRSF3(−164)-R	GTTCTTGCGGCCGCCTATTTCCTTTCATTTGACC

## Data Availability

The data presented in this study are available on request from the corresponding author.

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
