# Peer review of "Multiple Phosphorylations of SR Protein SRSF3 and Its Binding to m6A Reader YTHDC1 in Human Cells"

_cells, 2022, doi:10.3390/cells11091461_

Round 1

Reviewer 1 Report

The manuscript by Tatsuno and Ishigaki explores the contribution of phosphorylation of SRSF3 and its various domains in cellular localization, and its interaction with YTHDC1. Overall, the study is a good starting point towards understanding these issues. Additional interaction assays with the mutated derivatives may be helpful to build a more informative study. The writing needs to be improved to explain in a clearer fashion the results.

  1. In contrast to what is alluded to from the title (phosphorylation of YTHDC1), the manuscript is about the phosphorylation of SRSF3.
  2. 1. wild (wild-type?). GFP-SRSF3 (1-83) while existing in the cytoplasm strongly localizes to the nucleus. Same with GFP-SRSF3 (133-164). Therefore, this is not consistent with the statement that the “RRM or the latter half of RS domain might not have potentials for nuclear localization … ” (lane 170). May be its potential has been weakened but it is far from being absent.
  3. The sentence starting with “Although…” (lane 192) is not a full sentence.
  4. The rationale for interpreting the Phos-tag results (fig. 3) must be better explained. Since the full protein is used, if mutating one serine residue still promotes a shift like the wild-type protein, can one conclude that this residue is not normally phosphorylated? Does a lower migration imposed by a mutation indicate loss of phosphorylation (e.g. 108SA)? Is the contribution of each phosphorylation event on gel migration assumed to be the same? What does it mean when a mutation slows migration in the Phos-gel relative to the wild-type protein (e.g. 138SA): more phosphorylation in the rest of the protein? Is that expected?
  5. 4. I do not understand why position 124 is claimed to be phosphorylated since its mutation does not change migration (likewise for mutations 120 and 122). Since no mutation was made for 130, it should not be part of the region claimed to be phosphorylated.
  6. Last paragraph of page 6 should be rewritten as it is very repetitive.
  7. We are told that the 5460SA mutant (mutations at 154-160) has a hyperphosphorylated band shift (lanes 212-213) but a bit later (lanes 218-219) the opposite is claimed: hyperphosphorylation from replacement at positions 154-160 disappeared…! Clearly, it did not disappear when looking at Fig. 4. This confusion is repeated in the second paragraph of page 12. Overall, the section “phosphorylation of RS domain of SRSF3” needs to be rewritten, as well as the second paragraph on page 12 which is very hard to follow.
  8. Lane 245 (Fig. 5). I do not understand why the results implied an interaction of SRSF3 with the C-terminal region of YTHDC1 since only the full YTHDC1 protein is used in the IP assay at this stage.
  9. Lane 293. Fig. 7b should be Fig. 7c?
  10. 8. Please describe what is the 1-132SA mutant? Why was this mutant tested, why not others?
  11. Please indicate in the Result section what are the author’s conclusions from these IP (Figs. 7 and 8).
  12. The discussion has more text than the results. It should be more succinct. The interpretation of the results should be in the Results section which would help understanding each section sequentially. The remaining discussion needs considerable rewriting as the syntax currently compromises understanding. I was just completely lost trying to navigate it.

Author Response

  1. In contrast to what is alluded to from the title (phosphorylation of YTHDC1), the manuscript is about the phosphorylation of SRSF3.

Thank you for your comment. We changed the title.

  1. 1. wild (wild-type?). GFP-SRSF3 (1-83) while existing in the cytoplasm strongly localizes to the nucleus. Same with GFP-SRSF3 (133-164). Therefore, this is not consistent with the statement that the “RRM or the latter half of RS domain might not have potentials for nuclear localization … ” (lane 170). May be its potential has been weakened but it is far from being absent.

Thank you for your comments. We eliminated all of photos and changed the category of our manuscript from regular article to communication.

  1. The sentence starting with “Although…” (lane 192) is not a full sentence.

We changed to “Although the replaced alanine residues could not be phosphorylated (Figure 3b).”

  1. The rationale for interpreting the Phos-tag results (fig. 3) must be better explained. Since the full protein is used, if mutating one serine residue still promotes a shift like the wild-type protein, can one conclude that this residue is not normally phosphorylated? Does a lower migration imposed by a mutation indicate loss of phosphorylation (e.g. 108SA)? Is the contribution of each phosphorylation event on gel migration assumed to be the same? What does it mean when a mutation slows migration in the Phos-gel relative to the wild-type protein (e.g. 138SA): more phosphorylation in the rest of the protein? Is that expected?

  1. I do not understand why position 124 is claimed to be phosphorylated since its mutation does not change migration (likewise for mutations 120 and 122). Since no mutation was made for 130, it should not be part of the region claimed to be phosphorylated.

Thank you for your two important comments.

Compared to control, band shift difference of 138SA is caused by the different phosphorylation status of polypeptides. Cases such as 138SA probably affect phosphorylation around 108, 115, 124-130. As for the phosphorylation of 124 and 130, they were considered to be phosphorylated based on the results originally examined in the fragment. The phosphorylation of the 130th serine was band-shifted when the 128th serine was collapsed, suggesting full-length phosphorylation of SRSF1 (Ref35). As we believe was mentioned in the paper, the RS repeats closer to the RRM are phosphorylated successively from the C-terminus, and the frequency of phosphorylation is basically lower at the N-terminus. In fact, the 124th serine is phosphorylated less frequently than the 126th and 128th serines, and there is no doubt that there appears to be almost no difference in phosphorylation frequency at full length. However, it is not likely that it is not phosphorylated at all.

  1. Last paragraph of page 6 should be rewritten as it is very repetitive.

We deleted it.

  1. We are told that the 5460SA mutant (mutations at 154-160) has a hyperphosphorylated band shift (lanes 212-213) but a bit later (lanes 218-219) the opposite is claimed: hyperphosphorylation from replacement at positions 154-160 disappeared…! Clearly, it did not disappear when looking at Fig. 4. This confusion is repeated in the second paragraph of page 12. Overall, the section “phosphorylation of RS domain of SRSF3” needs to be rewritten, as well as the second paragraph on page 12 which is very hard to follow.

The 154-160SA mutant appears to be overphosphorylated because of increased phosphorylation of serines 108, 115, 124-130 (and possibly 120 and 122) near the N-terminus. In fact, the band shift disappears in MSA2, so we are not contradicting ourselves.

  1. Lane 245 (Fig. 5). I do not understand why the results implied an interaction of SRSF3 with the C-terminal region of YTHDC1 since only the full YTHDC1 protein is used in the IP assay at this stage.

We deleted this sentence.

  1. Lane 293. Fig. 7b should be Fig. 7c?

We changed b to c.

  1. Please describe what is the 1-132SA mutant? Why was this mutant tested, why not others?

The 1-132SA mutant is 108, 115 and 124-130SA mutant. We added “The 1-132SA mutant is 1-132 SRSF3 with 108, 115 and 124-130SA mutants.”. to manuscript. The reason why 1-132SA is listed is to see if phosphorylation affects the binding of YTHDC1 to SRSF3.

  1. Please indicate in the Result section what are the author’s conclusions from these IP (Figs. 7 and 8).

We added “This result shows the no relationship between phosphorylation and binding to YTHDC-1.” for Fig8.

  1. The discussion has more text than the results. It should be more succinct. The interpretation of the results should be in the Results section which would help understanding each section sequentially. The remaining discussion needs considerable rewriting as the syntax currently compromises understanding. I was just completely lost trying to navigate it.

We modified our manuscript and divided the discussion into some paragraphs. We also deleted the localization photos to make our conclusion clear.

Reviewer 2 Report

Greatly executed study. There are no major issues with study design or results that are presented. However, an important point to consider-

It is important for the authors to consider this comment- Please provide the high resolution full size images of the localization of mCherry-SRSF2 as nuclear speckles in Figure 2. Same rules apply for Figure 4 and 9. The phenotype must be presented with utmost clarity. Use 60 X or 100 X oil lens and Z-stack features to show the localization pattern of these speckles. Also co-stain with Nuclear laminin or etc, to authenticate these findings.

Author Response

Greatly executed study. There are no major issues with study design or results that are presented. However, an important point to consider-

It is important for the authors to consider this comment- Please provide the high resolution full size images of the localization of mCherry-SRSF2 as nuclear speckles in Figure 2. Same rules apply for Figure 4 and 9. The phenotype must be presented with utmost clarity. Use 60 X or 100 X oil lens and Z-stack features to show the localization pattern of these speckles. Also co-stain with Nuclear laminin or etc, to authenticate these findings.

Thank you for your kind comments. We eliminated all of photos and changed the category of our manuscript from regular article to communication.

Reviewer 3 Report

Summary:

Nuclear m6A reader, YTHDC1, interacts with hypophosphorylated SRSF3 to mediate the nuclear export of methylated mRNAs. The C-terminus of YHTHDC1 interacts with SRSF3 to regulate mRNA transport. However, the residues critical for phosphorylation of SRSF3 are unknown. In the manuscript by Tatsuno and Ishigaki, the authors aimed to identify the residues essential for phosphorylation of SRSF3 and domains required for the interaction between SRSF3 and YTHDC1. Using Co-immunoprecipitation and phos-tag gels, the authors revealed serine residues at 108, 115, and 124-128 needs to be dephosphorylated for the interaction with YTHDC1. Moreover, the authors showed that SRSF3 could interact with both C- and N-terminus of YTHDC1 and that the RRM domain of SRSF3 can interact with the C-terminus of YTHDC1. Although revealing residues required for interaction of SRFS3 with the YTHDC1 are important to the field, the manuscript in the current falls short and needs additional data to strengthen the conclusions. Specific comments are below.

Major Comments:

  1. Authors use localization studies and nuclear speckle formation to show the biological significance of the residues identified. Moreover, the authors themselves demonstrate that there is no correlation between the localization and phosphorylation status (as shown in figure 4C). So, to show the dephosphorylation of residues 108, 115, 124-128 is important for the interaction, the authors should show that serine to alanine mutations at these positions do impact mRNA transport. One way to address this is by carrying fluorescence in situ hybridization of poly-A mRNA and showing that mRNA transport is indeed affected (Please see PMID: 28984244).

Minor comments:

  1. The title is not appropriate; the authors did not show any data that revealed the effect on mRNA regulation. Please update the title.
  2. It appears that the mutation of serine to alanine at 140 and 148 seems to have more effect than serine at 124.
  3. In a previous manuscript PMID: 28984244, it is shown that SRSF3 interacts with the C-terminus of YTHDC1 but not with the N-terminus. The authors need to discuss this and also mention it in the results section.
  4. Figure 6 legend: it is written as wild; please correct it to wild type.
  5. In figures 5 and 6, immunoblots, both regular and phos-tag blots, show multiple bands. If the authors think they are nonspecific bands, it has to be mentioned in the figure legend.
  6. Immunoblots in figure 3b, especially 154SA to 160SA are challenging to see and have diffused bands. I suggest authors replace the image with a better one.
  7. The images where the authors show localization of various mutants are too small, and the authors need to enlarge the images so that speckles are visible.
  8. Figures 3, 4, 5, 6, 7, and 8, immunoblots should be marked for the size.

Author Response

Major Comments:

  1. Authors use localization studies and nuclear speckle formation to show the biological significance of the residues identified. Moreover, the authors themselves demonstrate that there is no correlation between the localization and phosphorylation status (as shown in figure 4C). So, to show the dephosphorylation of residues 108, 115, 124-128 is important for the interaction, the authors should show that serine to alanine mutations at these positions do impact mRNA transport. One way to address this is by carrying fluorescence in situ hybridization of poly-A mRNA and showing that mRNA transport is indeed affected (Please see PMID: 28984244).

Thank you for your valuable comment. We gave up this trial and changed the category of our manuscript from regular article to communication.

Minor comments:

  1. The title is not appropriate; the authors did not show any data that revealed the effect on mRNA regulation. Please update the title.

We changed the title to “Multiple phosphorylations and binding of SR protein SRSF3 to m6A reader YTHDC1 in human cells”.

  1. It appears that the mutation of serine to alanine at 140 and 148 seems to have more effect than serine at 124.

As shown in Figure 2, 126SA and 128SA were more effective than 124SA, but 124SA, 140SA, and 148SA were comparable to Wild.

  1. In a previous manuscript PMID: 28984244, it is shown that SRSF3 interacts with the C-terminus of YTHDC1 but not with the N-terminus. The authors need to discuss this and also mention it in the results section.

We added “This result differs from that of Roundtree et al. (23), i.e., that the N-terminus of YTHDC1 had no binding ability, which may be due to differences in experimental conditions such as buffer and the effect of the tag.” in result section.

  1. Figure 6 legend: it is written as wild; please correct it to wild type.

Thanks for the comment, we added the “type”.

  1. In figures 5 and 6, immunoblots, both regular and phos-tag blots, show multiple bands. If the authors think they are nonspecific bands, it has to be mentioned in the figure legend.

Thank you for pointing out. In each figure legend, we added the sentence "Some nonspecific bands are observed.”

  1. Immunoblots in figure 3b, especially 154SA to 160SA are challenging to see and have diffused bands. I suggest authors replace the image with a better one.

For this figure, we replaced it with an image with a longer exposure time.

  1. The images where the authors show localization of various mutants are too small, and the authors need to enlarge the images so that speckles are visible.

All photos have been deleted.

  1. Figures 3, 4, 5, 6, 7, and 8, immunoblots should be marked for the size.

Thank you for your comments. We added size mark.

Round 2

Reviewer 1 Report

"Although the replaced alanine residues could not be phosphorylated (Figure 2b)"  is still not a complete sentence. 

same with this sentence: "

Although we predicted that only the RRM region would co-immuno-precipitate with the C-terminal YTHDC1 fragment YTHDC1(493–727), which contained arginine-rich amino acid sequences, Myc-YTHDC1 also co-immunoprecipitated"

Shouldn't this sentence "The 1-132SA mutant 247 is 1-132 SRSF3 with 108, 115 and 124-130SA mutant" be "The 1-132SA mutant 247 is 1-132 SRSF3 with 108, 115 and 124-130SA mutations"

When the authors say "around 124-128" this could possibly mean form 122 to 130. Is that the case?

This sentence is poor English: "This result shows the no relationship between phosphorylation and binding to YTHDC-1"

Author Response

Thank you for your kind comments. We have revised the manuscript as follows.

(1) "Although the replaced alanine residues could not be phosphorylated 
(Figure 2b)"  is still not a complete sentence. 

We changed

“The SA mutants at position 108, 115, and around 124–128, produced lower band shifts in the phos-tag analysis (Figure 2a). The phosphorylation pattern of the SA mutants at position around 124–128 was similar to that of the N-terminal RS repeats of SRSF1 phosphorylation by SRPK1 [35]. However, many of the SA mutations, which reside in the latter half of RS domain, showed upper band shifts. Although the replaced alanine residues could not be phosphorylated (Figure 2b). In addition, the multiple band shifts show the phosphorylation of multiple serine in SRSF3 polypeptide.”

to

“From the phos-tag analysis (Figure 2a), results showed that SA mutants at positions 108, 115, and around 124–130, produced lower band shifts. We also observed that although the phosphorylation pattern of the SA mutants at positions around 124–130 was similar to that of the N-terminal SRSF1 phosphorylation RS repeats by SRPK1 [35], many SA mutations residing at the latter half of the RS domain showed upper band shifts. As a result, the replaced alanine residues could not be phosphorylated (Figure 2b). Additionally, multiple band shifts showed the phosphorylation of multiple serine residues in SRSF3 polypeptide.”.

(2) same with this sentence: "Although we predicted that only the RRM 
region would co-immuno-precipitate with the C-terminal YTHDC1 fragment 
YTHDC1(493–727), which contained arginine-rich amino acid sequences, Myc-YTHDC1 also co-immunoprecipitated"

We changed

“Therefore, we decided to examine SRSF3 binding with C-terminal fragments of YTHDC1 (Figure 5). Although we predicted that only the RRM region would co-immunoprecipitate with the C-terminal YTHDC1 fragment YTHDC1(493–727), which contained arginine-rich amino acid sequences, Myc-YTHDC1 also co-immunoprecipitated. These results implied that YTHDC1 may bind to both the RRM and RS domains of SRSF3 through C-terminal sequences.”

to

“Therefore, we examined SRSF3 binding with the C-terminal fragments of YTHDC1 (Figure 5). We predicted that even though only the RRM region coimmunoprecipitated with the C-terminal YTHDC1 fragment, YTHDC1(493–727) (containing arginine-rich amino acid sequences), Myc-YTHDC1 was also coimmunoprecipitated. These results imply that YTHDC1 was bound to the RRM and RS domains of SRSF3 through C-terminal sequences.”.

(3) Shouldn't this sentence "The 1-132SA mutant 247 is 1-132 SRSF3 with 
108, 115 and 124-130SA mutant" be "The 1-132SA mutant 247 is 1-132 SRSF3 
with 108, 115 and 124-130SA mutations"

We changed the sentence to proposed sentence. Thank you.

(4) When the authors say "around 124-128" this could possibly mean form 
122 to 130. Is that the case?

It could also be said to be 122-130.However, since 122 was not phosphorylated at all in the fragment experiment, let alone the full-length experiment, the phrase "around 124-128" refers more to 124-130 than to 122-130. Therefore we changed to 124-128 to 124-130.

(5) This sentence is poor English: "This result shows the no 
relationship between phosphorylation and binding to YTHDC-1"

We changed

“Additionally, YTHDC1(1–353) was confirmed to bind to SRSF3(1–132) SA mutant (Figure 7). This result shows the no relationship between phosphorylation and binding to YTHDC-1.”

to

“Investigations also confirmed the binding of YTHDC1(1–353) to the SRSF3(1–132) SA mutant (Figure 7). However, our results showed no relationship between phosphorylation and binding to YTHDC-1.”.

Reviewer 2 Report

Authors have made the modifications according to their experimental evidence and technical strengths. good job!

Author Response

Thank you very much.

Reviewer 3 Report

The authors answered all queries. I have no further comments.

Author Response

Thank you for your kind comments.